# Modeling collective behavior in groups of mice housed under semi-naturalistic conditions

**Xiaowen Chen[1], Maciej Winiarksi[2], Alicja Puścian[2], Ewelina Knapska[2], Thierry Mora[1]\*†, Aleksandra M Walczak[1]\*†**

[1]Laboratoire de physique de l'École normale supérieure, CNRS, PSL University, Sorbonne Université, Université Paris Cité, Paris, France; [2]Center of Excellence for Neural Plasticity and Brain Disorders, BRAINCITY, a Nencki-EMBL Partnership, Nencki Institute of Experimental Biology of Polish Academy of Sciences, Warsaw, Poland

**\*For correspondence:**
thierry.mora@phys.ens.fr (TM);
aleksandra.walczak@phys.ens.
fr (AMW)

†These authors contributed equally to this work

## eLife Assessment

This **valuable** work investigates the social interactions of mice living together in a system of multiple connected cages. It provides **solid** evidence for a statistical approach capturing changes in social interactions after manipulating prefrontal cortical plasticity. This research will be of broad interest to researchers studying animal social behavior.

**Abstract** In social behavior research, the focus often remains on animal dyads, limiting the understanding of complex interactions. Recent trends favor naturalistic setups, offering unique insights into intricate social behaviors. Social behavior stems from chance, individual preferences, and group dynamics, necessitating high-resolution quantitative measurements and statistical modeling. This study leverages the Eco-HAB system, an automated experimental setup that employs radiofrequency identification tracking to observe naturally formed mouse cohorts in a controlled yet naturalistic setting, and uses statistical inference models to decipher rules governing the collective dynamics of groups of 10–15 individuals. Applying maximum entropy models on the coarse-grained co-localization patterns of mice unveils social rules in mouse hordes, quantifying sociability through pairwise interactions within groups, the impact of individual versus social preferences, and the effects of considering interaction structures among three animals instead of two. Reproducing co-localization patterns of individual mice reveals stability over time, with the statistics of the inferred interaction strength capturing social structure. By separating interactions from individual preferences, the study demonstrates that altering neuronal plasticity in the prelimbic cortex – the brain structure crucial for sociability – does not eliminate signatures of social interactions, but makes the transmission of social information between mice more challenging. The study demonstrates how the joint probability distribution of the mice positions can be used to quantify sociability.

## Introduction

Social behavior is fundamental for numerous animal species, encompassing human societies. From the dynamic spectacle of Mexican waves in a football stadium to the intricate waggle dance of bees, the diverse manifestations of social interaction raise a pivotal question: How do these social behaviors come to fruition, and what roles do individuals play in their emergence?

In recent decades, the exploration of social behavior has predominantly centered around studying animal dyads in controlled laboratory conditions. However, these experimental paradigms inherently impose limitations on investigating intricate social behaviors that often involve more than two interacting individuals. Studies on social interactions frequently employ tests with brief observation periods, during which animals are evaluated in novel environments, accompanied by the presence of an experimenter that induces stress, influencing social behavior (*Allsop et al., 2014*; *Hurst and West, 2010*; *Sandi and Haller, 2015*; *Chesler et al., 2002*; *Sorge et al., 2014*). Recently, there has been a notable shift toward conducting experiments in natural settings, involving animal groups such as flocks of birds (*Ballerini et al., 2008*) or swarms of midges (*Attanasi et al., 2014*). Additionally, seminaturalistic environments, exemplified by fish in tanks (*Katz et al., 2011*; *Herbert-Read et al., 2011*; *Gautrais et al., 2012*), marching locusts in arenas (*Buhl et al., 2006*), flocks of sheep (*Ginelli et al., 2015*), and hordes of rodents (*Kondrakiewicz et al., 2019*; *Puścian et al., 2016*), are increasingly being utilized. These approaches present unique opportunities for the comprehensive quantification of complex social interactions and sociability.

Mice stand out as a valuable model system for delving into the complexities of social behavior, given their intricate manifestation of various social behaviors. They tend to form cohesive groups, showcasing both amicable and agonistic behaviors. Depending on the environmental context, mice demonstrate territoriality and dynamic social hierarchies (*Williamson et al., 2016*). Communication among mice is extensive, primarily mediated through odors, allowing them to convey emotional states such as stress, fear, and preferences in food (*Jeon et al., 2010*; *Galef, 2002*). Additionally, mice exhibit prosocial behaviors, actively assisting distressed fellow mice in need (*Gonzalez-Liencres et al., 2014*). Decades of research have extensively explored social interactions between pairs of mice, while the study of mouse groups has only recently become feasible with advancements in high-throughput technologies, particularly radiofrequency identification (RFID) (*Freund et al., 2013*; *Puścian et al., 2016*). The Eco-HAB system, utilized in this study, leverages RFID tracking to observe naturally formed cohorts of mice in a controlled yet naturalistic environment, enabling longitudinal experiments on sociability with minimal human interference (*Puścian et al., 2016*).

Social behavior arises from a combination of chance, individual preferences, group structure, and the transmission of preferences and interactions among group members. To unravel these elements and understand the establishment of social networks and hierarchies, we need not only high-resolution quantitative measurements of behavior over extended periods, but also statistical modeling to construct interaction models of collective behavior. One particular statistical method that has been successfully applied to identify interaction models in a diverse range of biological networks are maximum entropy models. Among many examples, maximum entropy models have successfully explained social rules governing collective behavior in bird flocks and mouse hordes (*Bialek et al., 2012*; *Shemesh et al., 2013*). These models help distinguish observed correlations, like the clustering of mice in a specific location, from direct interactions or individual preferences. *Shemesh et al., 2013* pioneered the use of these models in studying mouse group behavior, revealing the significance of higher-order interactions in co-localization patterns. While (*Shemesh et al., 2013*) utilized video tracking of groups of four mice, our Eco-HAB setup employs RFID technology for tracking groups of 10–15 mice, providing more compact data with longer recording times but lower spatial resolution. In this study, we integrate Eco-HAB recordings with statistical inference to construct models of collective behavior, focusing on the statistics of system states to identify interaction structures within the group. Our focus is on quantifying sociability in mouse hordes through the inferred interactions within groups, ensuring statistical power. We explore whether interactions between pairs can explain collective behavior and examine how social structure evolves over time. We analyze the effects of individual versus social preferences and investigate the impact of considering three animals instead of two. Using a data analysis approach based on wild type C57BL/6J male and female mice, we discuss social structure and sociability changes in mice with temporary prefrontal cortex (PFC) plasticity modification.

Furthermore, we explore whether interventions of brain regions that are crucial for processing social information change the interaction patterns among individuals in the Eco-HAB. The PFC plays a crucial role in processing social information, understanding others' emotions, maintaining social hierarchy, and transmitting information about food safety in both rodents and humans (*Bicks et al., 2015*; *Demolliens et al., 2017*). Neuronal activity of the PFC is correlated with proximity to conspecifics,

and studies in mice reveal distinct PFC responses to social and non-social olfactory stimuli (*Lee et al., 2016*; *Levy et al., 2019*). The PFC integrates existing knowledge with new information about self and others, demonstrating dynamic neuronal plasticity (*Denny et al., 2012*). In cognitive tasks involving the PFC and subcortical areas, neuronal connectivity refines more rapidly in the former, highlighting its adaptability to changing environments (*Yizhar and Levy, 2021*; *Loureiro et al., 2019*). Tissue inhibitors of metalloproteinases (TIMPs), particularly TIMP-1, influence synaptic plasticity by inhibiting matrix metalloproteinases (MMPs), especially MMP-9 (*Attwood et al., 2011*; *Bach et al., 2018*; *Vaillant et al., 1999*; *Ould-yahoui et al., 2009*; *Dziembowska and Wlodarczyk, 2012*). TIMP-1 is involved in long-term potentiation (LTP), a crucial process for cellular-level memory formation (*Gorkiewicz et al., 2015*; *Okulski et al., 2007*). This sustained release impedes the updating of neuronal connectivity in the prelimbic part of the PFC (PL), crucial for maintaining social structure (*Wang et al., 2011*; *Bicks et al., 2021*; *Wang et al., 2014*). Our study employs nanoparticles for gradual TIMP-1 release over several days (*Chaturvedi et al., 2014*) to impact plasticity in the PL on the changes in group behavior.

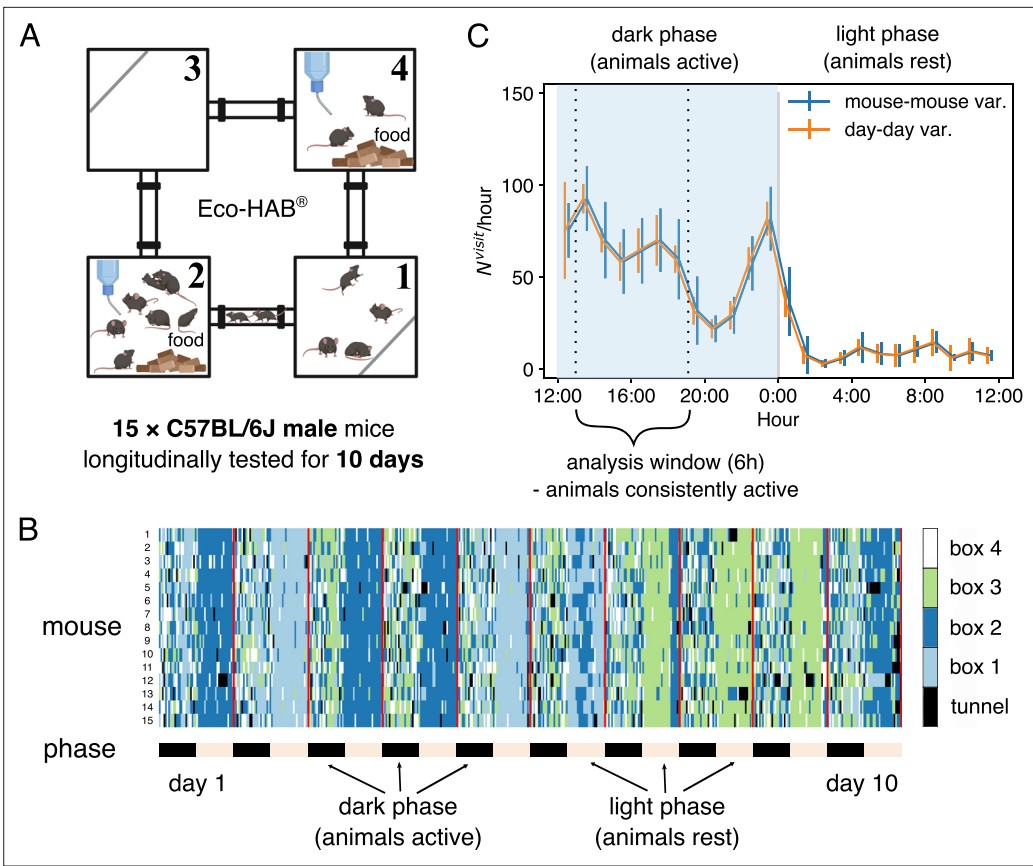

**Figure 1.** Mice were tested in Eco-HAB, a system for automated, ecologically relevant assessment of voluntary behavior in groups of mice. Animals were tested for 10 days. (**A**) Schematic of the Eco-HAB system, where four compartments are connected with tunnels. Food and water are available ad libitum in compartments 2 and 4. (**B**) Time series of the location of 15 mice over 10 days, as aligned to the daylight cycle. (**C**) Circadian clock affects the activity of the mice, measured by the number of transitions in each hour averaged over the 15 mice and the 10 days. Error bars represent SD across all mice (mouse-mouse variability, in blue) or across all days for the mean activity level for all mice (day–day variability, in orange). The two curves are slightly shifted horizontally for clearer visualization. We focus the following analysis on the data collected during the first half of the dark phase, between 13:00 and 19:00 (shaded region).

The online version of this article includes the following figure supplement(s) for figure 1:

**Figure supplement 1.** Stability of the data, given by the time evolution across 10 days of the experiment and the scatter plot between the observables measured using the first 5 days of the data versus the last 5 days of the data for cohort M1 ($N = 15$).

# Results

## Recording of mice location in naturalistic environment

Eco-HAB is an automated, ethologically relevant experimental apparatus that tracts voluntary behavior in group-housed mice (**Puścian et al., 2016**). Constructed to simulate notable characteristics of natural murine environment, it consists of four connected large compartments, two of which contain food and water (**Figure 1A**). Cohorts of 10–15 mice are introduced into the Eco-HAB, where they behave freely while their locations are tracked over time. The details of used mouse strains and cohorts' compositions can be found in the 'Materials and methods'. The compartments are connected with tube-shaped corridors resembling underground tunnels, on whose ends there are 125 kHz antennas recording every time a mouse crosses with an accuracy of over 20 Hz. Each mouse is tagged with a unique RFID tag. The mice are recorded for 10 days with alternating 12-hour-long light–dark phases that simulate the day–night cycle.

The location of each mouse at each time is reconstructed using the time stamps, reducing the data to a discrete time series, $\sigma_t$ at time $t = 1, 2, \ldots, T$, with possible values of the locations $\sigma_t = 1, 2, 3, 4$ corresponding to the four compartments. The time resolution for the discretization is set to 2 seconds. As shown by the color-coded location traces in **Figure 1B**, the majority of mice are often found in the same compartment, especially in the non-active light phases: this corresponds to the ethological behavior – mice tend to sleep in a pile to keep each other warm (see **Rydzanicz et al., 2024**). This suggests that the behavior depends on latent variables, that is, the circadian clock.

To ensure the relative consistency of the analyzed data, we used the observed rate of transitions to measure the activity of each mouse, and choose an analysis window of 6 hours covering the first half of the dark phase (13:00–19:00), which corresponds to the most active time on each day for the entire duration of the experiment. The variability of activity across individuals is larger than the day-to-day variability for a single mouse, suggesting that the level of locomotor activity is a well-defined individual characteristic (**Figure 1C**).

## Pairwise interaction model explains the statistics of social behavior

We first establish a quantification of sociability by building probabilistic interaction models for groups of mice. Following previous work (**Puścian et al., 2016**), we use the in-cohort sociability, which measures the excess probability of two mice being found in the same compartment compared to the case where they are independent. Mathematically, in-cohort sociability is defined as

$$C_{ij}^{\text{data}} = \sum_{r=1}^{k} \left( f_{ij}(r, r) - f_i(r) f_j(r) \right) \tag{1}$$

where $f_i(r) \equiv m_{ir}^{\text{data}}$ and $f_{ij}(r, r')$ are respectively the empirical frequencies of finding a mouse $i$ in compartment $r$, and a pair of mice $(i, j)$ in compartments $r, r'$.

As schematically explained in **Figure 2A**, in-cohort sociability is due to pairwise interactions between each pair of mice and modifies how likely they are to be found in each compartment with respect to the mice's innate preference for that compartment. However, considering the presence of more than two animals, in-cohort sociability is not an effective measure of social structure of the group: two animals with zero attraction to one another can still be found to have a high in-cohort sociability, if a third animal has a strong social bond with both of them, since they all will be spending time with one another.

Since measurements of location preference, and the in-cohort sociability, together with the dynamic observables such as the rate of activity, are stable over time (**Figure 1—figure supplement 1**), it invites a quantitative modeling of the joint-probability distribution of the co-localization of mice.

To distinguish social structure interactions from the effective correlations that define in-cohort sociability, we build a maximum entropy model with pairwise interactions. This approach constrains the joint probability distribution of all the possible co-localization patterns of all mice to reproduce the empirical occupation frequencies $m_{ir}^{\text{data}}$ and the in-cohort sociability $C_{ij}^{\text{data}}$, while otherwise remaining as random as possible (**Jaynes, 1957**; **Schneidman et al., 2006**; **Shemesh et al., 2013**). With these assumptions, the joint probability distribution of the mice co-localization patterns can be written as

$$P^{(2)}(\sigma) \propto e^{\sum_{i,r} h_{ir} \delta_{\sigma_i, r} + \sum_{i<j} J_{ij} \delta_{\sigma_i, \sigma_j}}, \tag{2}$$

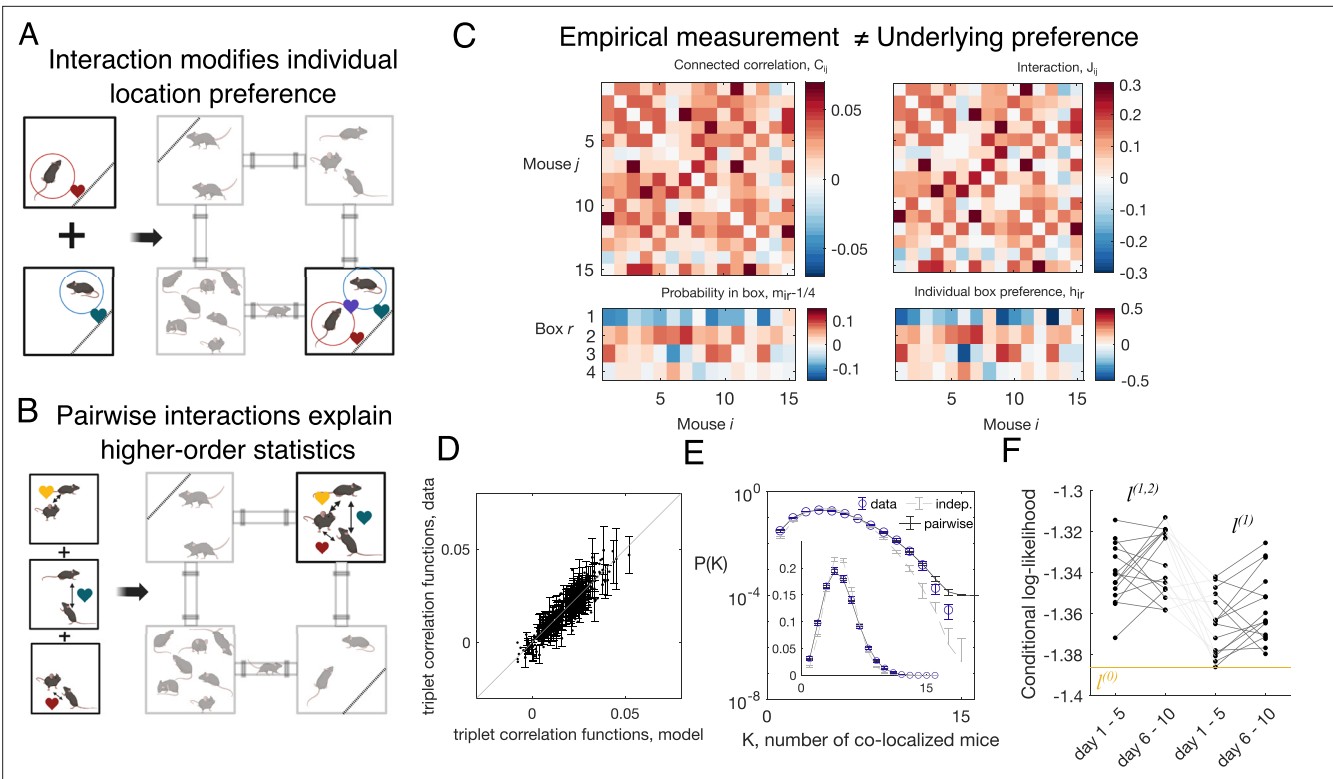

**Figure 2.** Mice in Eco-HAB interact pairwisely. (**A, B**) The schematics showing pairwise interactions: two mice are more likely to be found in the same compartment than the sum of their individual preference implies (**A**); also, the probability for three mice being in the same compartment can be predicted from the pairwise interactions (**B**). (**C**) From pairwise correlation $C_{ij}$, defined as the probability for mouse $i$ and mouse $j$ being in the same compartment (subtracted by the prediction of the independent model), and the probability for mouse $i$ to be found in compartment $r$ (subtracted by the model where each mouse spends equal amount of time in each of the four compartments), $m_{ir} - 1/4$, pairwise maximum entropy model learns the interaction strength between a pair of mice, $J_{ij}$, and the local field $h_i^r$, which gives the tendency for each mouse in each compartment. The data shown is the aggregated 5-day data from day 1 to day 5 of the C57BL/6J males (cohort M1, $N = 15$). The pairwise maximum entropy model can predict higher-order statistical structures of the data (schematics in **B**), such as the probability for triplets of mice being in the same compartment (subtracted by the prediction of the independent model, mathematically $f_{ijk}(r,r,r) - f_i(r)f_j(r)f_j(r)$) (**D**), and the probability of $K$ mice being found in the same compartment (**E**). Error bars in (**D**) and (**E**) are extrapolated from random halves of the data (see 'Materials and methods' for details). (**F**) Conditional log-likelihood of mouse locations, predicted by the pairwise model ($l^{(1,2)}$), the independent model ($l^{(1)}$), and the null model assuming no compartment preference or interactions ($l^{(0)}$, the yellow line), for each mice.

The online version of this article includes the following figure supplement(s) for figure 2:

**Figure supplement 1.** The pairwise maximum entropy model reproduces the probability for each mouse in each compartment, $m_{ir}$, and the probability for pairs of mice in the same compartment, $C_{ij}$, as given by the data.

**Figure supplement 2.** Learned parameters in the pairwise interaction model versus the observed statistics, plotted for the 5-day aggregate data from the first 5 days of the experiment on male cohort M1 before TIMP-1 treatment.

**Figure supplement 3.** The probability of $K$ mice found in the same compartment, predicted by the pairwise maximum entropy model, the independent model, and computed from the 5-day aggregate data for the first 5 days in male cohort M1 before TIMP-1 treatment ($N = 15$).

**Figure supplement 4.** Model-predicted in-state probability matches data observation for the aggregate data of first 5 days of experiment in mice cohort M1 – C57BL/6J male mice ($N = 15$), which shows the prediction of the inferred pairwise model is unbiased.

**Figure supplement 5.** Cross-validation for maximum entropy models with triplet interactions and models with pairwise interactions on combined 10-day data (cohort M1, $N = 15$).

**Figure supplement 6.** Cross-validation for maximum entropy models with pairwise interactions on combined data from a total of $K$ days (cohort M1, $N = 15$).

**Figure supplement 7.** Temporal consistency of inferred parameters from 5-day accumulated data for mice cohorts M1 ($N = 15$), M2 ($N = 13$), M3 ($N = 10$), M4($N = 12$).

**Figure supplement 8.** The conditional log likelihood is different for each cohort of C57BL/6J male mice $N = 13$ in cohort M2, $N = 10$ in cohort M3, and $N = 12$ in cohort M4 (before BSA injection), exhibiting individuality.

where $h_{ir}$ is the individual preference of mouse $i$ to be in compartment $r$, and $J_{ij}$ is the interaction between mouse $i$ and mouse $j$. The set of parameters $(\{h_{ir}, J_{ij}\})$ is learned through gradient descent (see *Figure 2—figure supplement 1* and 'Materials and methods' for details). The interactions $J_{ij}$ may be positive or negative. We see that although the structure of the interactions $J_{ij}$ follows that of in-cohort sociability $C_{ij}$, they are not identical (*Figure 2—figure supplement 2*). Likewise, individual mice preferences $h_{ir}$ are not equal to the occupation probability $m_{ir}$ (*Figure 2C*). Thus, this approach allows us to distinguish direct interactions from indirect ones.

To validate the model, we tested that it is able to predict higher-order features of the data, such as the probability of a specific combination of triplets of mice being in the same compartment (*Figure 2BD*), and the probability of observing $K$ mice in the same compartment (*Figure 2E*), with the overestimation at large $K$ possibly due to the limit of finite data. Although the model assumes the strength of interaction does not depend on which compartment the mice are in, our minimal model can predict probability of $K$ mice in certain compartments (*Figure 2—figure supplement 3*, compartments 1 and 3). We call in-state probability the distribution of box occupancy of each mouse given the position of all other mice (see 'Materials and methods'). The model prediction for in-state probabilities matches the observed one, showing that the model gives an unbiased estimate of individual mouse positions (*Figure 2—figure supplement 4*). Moreover, models with triplet interactions show signs of overfitting under cross-validation, which is mitigated when the triplet interactions are suppressed close to zero using L2 regularization (see 'Materials and methods' and *Figure 2—figure supplement 5*). These results show that pairwise interaction among mice are sufficient to assess the observed collective behavior.

## Choosing timescales for analysis

To construct an interaction model based on the steady-state distribution, we first need to consider the proper timescales for which we average over the observables, that is, mean and correlation of the mice co-localization patterns. If the timescale is too short, then the error of estimation may be large. More severely, the system may not have enough time to equilibrate, and the time average will not result in the steady-state distribution. On the other hand, if the timescale is too long, we lose biologically meaningful information about the temporal evolution of the system, such as the adaptation of the mice in a new environment and evolution of the social interaction strength.

To identify the proper timescale, we systematically conduct cross-validation for pairwise maximum entropy models constructed using $K$ days of data, where $K = 1, 2, 3, 4, 5, 10$, and each day of the data contains the 6 hours when the mice are most active. The data is then separated into training sets that consist of 5 hours of the data from each day and test sets that consist of 1 hour of the data each day. Pairwise maximum entropy models with L2 regularization with strength $\beta_J$ imposed on the pairwise interactions are learned from the training set, and the training- and the test-set likelihoods are computed. As shown in *Figure 2—figure supplement 6*, the test-set likelihood decreases as the regularization strength increases for cumulate data with number of days $K \geq 4$, indicating that the pairwise model generalizes well. We choose $K = 5$ for subsequent analysis as it does not overfit, and it gives us temporal information about how the interaction structures may change over the 10-day experiment.

## Stability of sociability over time

The data-driven model and its inferred parameters allow us to explore various aspects of social behavior. As the models are built using accumulated data from 5 days in the 10-day experiment, we first assess the temporal consistency of the chamber preference $h_{ir}$ and the inferred interaction parameters $J_{ij}$ of the four cohorts of C57BL/6J male mice (see *Table 1*), M1 ($N = 15$), M2 ($N = 13$), M3 ($N = 10$), and M4 ($N = 12$, before BSA injection).

As shown in *Figure 2—figure supplement 7*, for all four cohorts, the box preference tendency is consistent, with a Pearson's correlation coefficient between models learned from the first and the last 5 days being $0.6 \pm 0.1$. For cohorts M1, M3, and M4, the distribution of the individual chamber preferences is consistent over time, with the two-sample $F$-test for equal variance being non-significant, and the two-sample $t$-test for equal mean being either non-significant or with a p-value of 0.04 for cohort M1. For cohort M2, both the mean and the variance of the chamber preferences significantly changed between the first and the last 5 days of the experiment. Nonetheless, the Pearson's correlation

**Table 1.** Summary of experiments used in this study.

The column $N_{\mathrm{mice}}$ gives the number of mice in the cohort used for the analysis, with cohort M4 and F1 containing dead or inactive mice after injection. The original number of mice is included in the parenthesis, and the exclusion procedure is described in 'Exclude inactive and dead mice from analysis'. The column 'NP' indicates the load of the injected nanoparticles. The column 'Day 1' indicates the first day of observation in each of the 10-day experiment.

| Cohort ID | Gender | $N_{\mathrm{mice}}$ | $T_{\mathrm{days}}$ | Day 1, WT | NP | Day 1, NP |
|---|---|---|---|---|---|---|
| M1 | Male | 15 | 10×2 | 180,511 | TIMP-1 | 180,526 |
| M2 | Male | 13 | 10 | 181,012 | N/A | N/A |
| M3 | Male | 10 | 10 | 200,701 | N/A | N/A |
| M4 | Male | 9 (out of 12) | 10×2 | 200,713 | BSA | 200,727 |
| F1 | Female | 13 (out of 14) | 10×2 | 180,413 | TIMP-1 | 180,430 |

between the two data segments remains large at p=0.57, indicating a consistency over time for the same cohort. Different cohorts exhibit different distributions of chamber preferences.

For the inferred interaction parameters $J_{ij}$, the distribution is consistent between the first 5 days and the last 5 days of the data. Specifically, for all four cohorts, the standard deviations of the interactions do not change between the first 5 days and the last 5 days of the data, as shown by two-sample $F$-test for equal variance. The mean of the inferred interactions does not change for cohorts M1 and M3; however, for cohorts M2 and M4, two-sample $t$-test for equal mean returns a p = 0.0034 and p = 0.0073, respectively, for the interactions. Notably, across from the four different cohorts, two-sample $F$-test with Bonferroni correction shows that the variance of all eight 5-day modeling is not significantly different. In contrast, cross-cohort comparison between male (cohort M1 before drug injection) and female (cohort F1 before drug injection) shows significantly different variance (p < 0.01 between the last 5 days of F1 and the first 5 days or the last 5 days of M1) and mean (p < 0.001) of the inferred interaction strength, which shows that these measures of sociability can be used to distinguish strains or genders. Nonetheless, the strengths of the individual interactions in the specific pairs of mice $i$ and $j$, $J_{ij}$, vary more notably, as given by Pearson's correlation coefficient $0.015 \pm 0.165$ (see *Figure 2—figure supplement 7C*). This implies that the maximum entropy model does not infer a social structure that is stable over time.

## Quantifying the influence of social versus individual preferences

Further, we ask how important social interactions are for determining mice behavior by measuring how much the data can be explained by the individual preferences for specific spaces within the territory versus the interactions with other mice. Mice are social animals, yet they perform many behaviors based on their individual moment-to-moment needs, and it is unclear a priori how much the social interactions influence mice behavior in comparison to their individual preference.

For each mouse $i$, we consider three nested models with increasing descriptive power: first, the null model assuming each mouse has the same probability of being found in each compartment, $P^{(0)}(\sigma_i) = 1/4$; second, the independent model that assumes no interactions among mice, and the probability of finding each mouse in each compartment is solely determined by their individual preferences, $P^{(1)}(\sigma_i) = f_i(\sigma_i)$; third, the inferred pairwise interaction model based on voluntarily spending time with other mice considered, $P^{(2)}(\sigma_i|\{\sigma_{j\neq i}\})$, using *Equation 2*.

We then quantified how well each model explains the data by comparing the mean log-likelihoods of finding a mouse in a given compartment, conditioned on the location of all other mice. As shown in *Figure 2F*, including information on pairwise interactions increases the log-likelihood of the data by as much as including information on individual compartment preferences, as shown by the similar values of the probability ratios $P^{(2)}/P^{(1)}$ and $P^{(1)}/P^{(0)}$. The likelihood is consistent between the first 5 days and the last 5 days of the experiment, but exhibits variability across different cohorts of animals within the same strain (*Figure 2—figure supplement 8*).

Another possible measure of sociability is the mutual information between a single mouse's location within the territory and the location of the rest of the cohort, which tells us how accurately the position of a single mouse can be predicted if the positions of all other mice are known (see details

in 'Materials and methods'). The possible values of the mutual information are between 0 and 2 bits, where 0 bits means no predictability, and 2 bits means perfect predictability. In our Eco-HAB data, the average mutual information for each mouse is $0.0323 \pm 0.0151$ (SD) bits for cohort M1, with the largest value being 0.06 bits, indicating that despite non-zero sociability the precise mouse position at any single moment is difficult to predict from the network.

## Effect of impairing neuronal plasticity in the PL on subterritory preferences and sociability

As a next step, we investigate the effects of impairing neuronal plasticity in the prelimbic cortex (PL), the brain structure containing neural circuits indispensable for both maintaining proper social interactions and encoding individual preferences (*Lee et al., 2016*). To that end, we inject animals with a Tissue Inhibitor of MetalloProteinases (TIMP-1), an enzyme regulating the activity of synaptic plasticity proteins. Changing its physiological levels was previously shown to disrupt the neuronal plasticity in various brain structures (*Knapska et al., 2013*; *Puścian et al., 2022*), including the PFC (*Okulski et al., 2007*) where an overexpression of TIMP-1 is found to block the activity of matrix metallo-proteinases and prevent the induction of late LTP in vivo. More specifically to the PL, it has been recently demonstrated in the Eco-HAB that injecting nanoparticles (NPs) gradually releasing TIMP-1 (NP-TIMP-1) (*Chaturvedi et al., 2014*) in the PL can reduce the mice's interest in chasing other animals (a proxy for their social ranks) and diminish persistence in seeking reward related to social olfactory cues (*Winiarski et al., 2025*), which supports the idea that TIMP-1 has regional-specific effects on behavioral processes.

Here, we measure the behavior of a cohort of $N = 15$ C57BL/6J male mice before and after the injection of NP-TIMP-1. A cohort of mice is introduced into the Eco-HAB, and their free behavior is measured for 10 days (see schematics in *Figure 3A*). Then, neuronal activity in the PL of the subjects is impaired by injecting nanoparticles releasing TIMP-1 into the PL. After recovery, animals are reintroduced into the Eco-HAB, and their behavior is measured for another 10 days. As a control, we also have a cohort (male cohort M4, $N = 9$) that is injected with nanoparticles loaded with bovine serum albumin, a physiologically neutral substance having no impact on neuronal plasticity (BSA, vehicle). The detailed experimental procedure can be found in *Winiarski et al., 2025*. To provide a perspective on both sexes, a female cohort is also included in this study (female cohort F1, $N = 13$); it was processed identically to the experimental group of males described above. For each 5 days of the experiment, we infer a pairwise model (*Equation 2*) and study the changes in the inferred interactions, as well as individual preferences for specific spaces within the territory. The choice of 5-day aggregated data for analysis is in line both with the proper timescales needed for the pairwise maximum entropy model to not overfit, and with the literature that TIMP-1 release from the TIMP-1-loaded nanoparticles is stable for 7–10 days after injection (*Chaturvedi et al., 2014*) (i.e., 2–5 days after the mice are reintroduced to Eco-HAB).

We can assess the change in both the interaction strength $J_{ij}$ and the individual preferences for compartments containing food $\Delta h_i$ following the prolonged release of TIMP-1. The individual preferences for compartment containing food show an increase in both its mean and its variance across all mice following treatment (*Figure 3C*), with a return to pretreatment levels after 5 days, consistent with the time course of TIMP-1 release (*Chaturvedi et al., 2014*). In comparison, the control cohort M4 shows an increase in preference for the compartments containing food after injection of the BSA-loaded nanoparticles, which does not return to base level after 5 days (*Figure 3H*). For the interaction strength, as shown by *Figure 3B*, the variability of interactions $J_{ij}$ is not significantly different before and after the TIMP-1 injection for the male cohort M1 (two-samples $F$-test for equal variance with Bonferroni correction; see 'Materials and methods'), although there exist a few outliers with strong interactions, which again returns to base level after 5 days of TIMP-1 injection. The increase in interaction variability is significant in the female cohort when comparing the first 5 days after treatment with the first 5 days before treatment (*Figure 3—figure supplement 1*), and is not observed for the male BSA cohort M4 (*Figure 3G*). We can also ask how TIMP-1-induced modification of PL plasticity affects individual mice by comparing the pairwise-specific interactions $J_{ij}$ before and after drug treatment. However, we cannot conclude much as Pearson's correlation coefficient between $J_{ij}$ shows almost no significant correlation across the four time periods in the above datasets for both the TIMP-1-treated cohort and the BSA-treated control cohort (*Figure 3—figure supplement 2*).

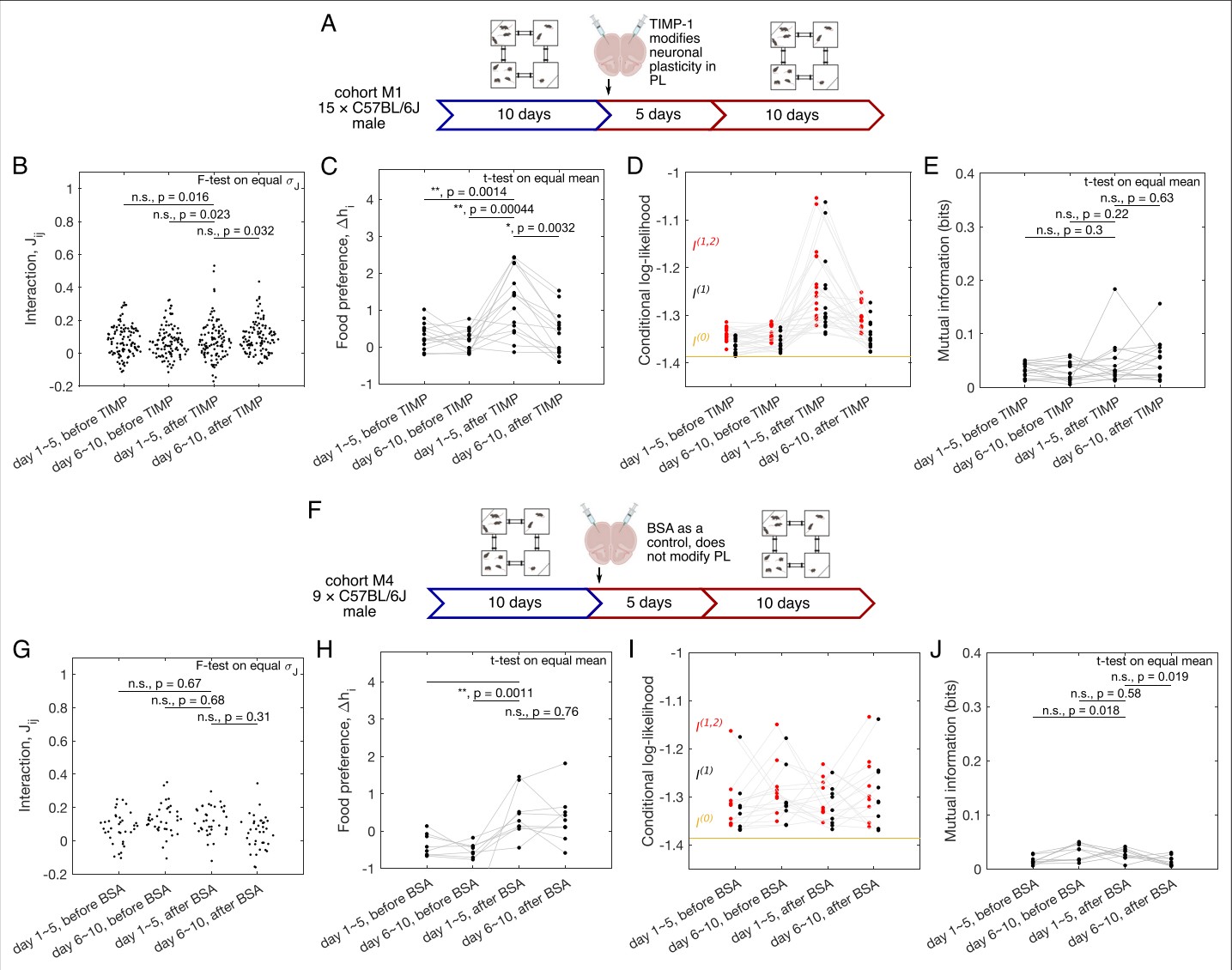

**Figure 3.** Quantification of sociability and the impact of the impaired neuronal plasticity in the prelimbic cortex (PL). (**A**) Schematic of the experiment, in which neuronal plasticity in the PL of the tested subjects was impaired with TIMP-1 treatment. A cohort of C57BL/6J male mice ($N = 15$) was tested in Eco-HAB for 10 days, and then removed from the cages for neuronal plasticity manipulation procedures. After a recovery period, they were placed back in Eco-HAB for another 10 days. For each of the 5-day aggregate of the experiment, both before and after TIMP-1 treatment, we plot (**B**) the model-inferred interactions $J_{ij}$, (**C**) preference for food compartments $\Delta h_i$, (**D**) conditional log-likelihood for the pairwise model, $l^{(1,2)}$, the independent model, $l^{(1)}$, and the baseline null model, $l^{(0)}$, and (**E**) mutual information between single mouse position and the rest of the network given by the inferred pairwise model. (**F–J**) Same as (**A–E**), now for male C57BL/6J mice subject to injection of BSA-infused nanoparticles, a control which does not impair neuronal plasticity in the PL (cohort M4, $N = 9$).

The online version of this article includes the following figure supplement(s) for figure 3:

**Figure supplement 1.** Same as *Figure 3A–E*, now for female C57BL/6J mice subject to injection of TIMP-1-infused nanoparticles (cohort F1, $N = 13$).

**Figure supplement 2.** Pearson's correlation coefficient between inferred interaction $J_{ij}$ from different 5-day aggregated data, before and after drug injection, for cohorts M1 (N = 15), M4 (N = 9), and F1 (N = 13).

To quantify the sociability of the entire cohort, we compute conditional likelihoods as introduced earlier as it measures how much the pairwise model explains the observed data compared to a model where mice behave independently. *Figure 3D* shows that for cohort M1 the model's likelihoods sharply increase following treatment, meaning that the behavior is more predictable. Represented by the independent model, the individual compartment preferences explain most of this increase, suggesting that TIMP-1 treatment reorganizes preferences for specific subterritories.

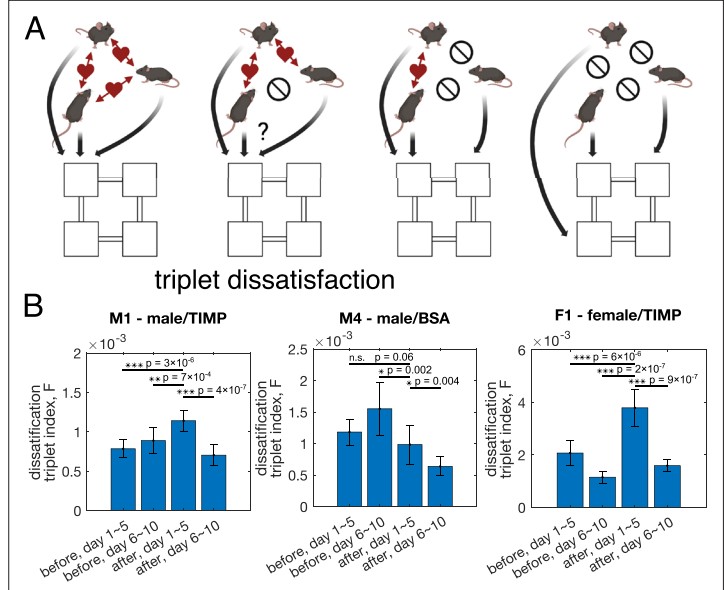

**Figure 4.** Effect of TIMP-1 on the structure of the interaction network. (**A**) Schematics of how triplets of mice may enter a state of 'dissatisfaction' due to competitive pairwise interactions. Dissatisfaction reduces the space of preferable states due to competitive interactions. (**B**) The global dissatisfaction triplet index (DTI), $F$, computed using inferred interaction from 5-day segments of the data shows that for both male and female mice treated with TIMP-1 the global DTI is significantly increased after drug treatment, for cohorts M1 ($N = 15$), M4 ($N = 9$), and F1 ($N = 13$). Two-sided Welch's $t$-test is performed to test the significance of the difference of the global DTI between the first 5 days after drug injection against the other 5-day segments of the data. Error bars estimate the data variability, which is generated by taking random halves of the data (see 'Materials and methods' for details).

The online version of this article includes the following figure supplement(s) for figure 4:

**Figure supplement 1.** Dissatisfaction triplet index $F$ computed using subgroups of mice for cohorts M1 ($N = 15$, $N_{\text{sub}} = 9$) and F1 ($N = 13$, $N_{\text{sub}} = 9$).

**Figure supplement 2.** Dissatisfaction triplet index computed using subgroups of mice from cohorts M1 and F1.

**Figure supplement 3.** The global dissatisfaction triplet index (DTI) computed using shuffled interaction, $F_{\text{shuffled}}$ versus the global DTI computed using the inferred interaction, $F_{\text{random half}}$, for cohorts M1 ($N = 15$), M4 ($N = 9$), and F1 ($N = 13$).

These differences decay back to pretreatment levels after 5 days, following the time course of drug release. A slightly smaller increase in model's likelihood is observed in the control cohort M4 after BSA injection (**Figure 3I**), suggesting that at least part of the change in compartment preferences can be due to the injection procedure rather than change in the neuronal plasticity itself. In contrast, the increasing model likelihood is not found in the female cohort F1, where the conditional likelihood remains constant after TIMP-1 treatment. However, the contribution of the pairwise interaction is increased (**Figure 3—figure supplement 1E**), which points to a sex specificity of observed effects.

This observation is further confirmed by the sociability measure of mutual information between single mouse location and the positions of the rest of the cohort, which was introduced earlier. The mutual information either does not change (for the male cohort M1, **Figure 3E**) or increases (for the female cohort F1, **Figure 3—figure supplement 1**) after the injection of TIMP-1.

## Impaired neuronal plasticity in the PL affects the structure of social interactions

The increasing variability of pairwise interactions and the non-decreasing mutual information between single mouse location and the location of the rest of the group upon TIMP-1 requires further investigation in the face of previous results showing that injecting TIMP-1 to the PL of wild type animals reduces their sociability. Thus, we examined the detailed group structure of pairwise interactions. We define the dissatisfaction triplet index (DTI) among a triplet of mice as $F_{ijk} \equiv -J_{ij}J_{jk}J_{ki}$ if and only if among the three pairwise interactions among mice $i$, $j$, and $k$, exactly one of them is negative (see **Figure 4A**

for schematics), and otherwise zero. Notice that DTI is analogous to the concept of 'frustration' in the physics of disordered systems. A positive DTI means a triplet of pairwise interactions where all of them cannot be satisfied simultaneously – for instance, if mouse $i$ likes to be with $j$ and $k$, but $j$ and $k$ do not like to be together. We define the global DTI by averaging the local DTI's across all possible triplets of mice. The larger the global DTI is for a cohort, the more difficult it is for the cohort to form cliques with multiple mice where the interactions among each possible pairs are positive, which may suggest possible difficulty in transmitting information between mice.

As shown in *Figure 4B*, PL-targeted plasticity disruption with TIMP-1 significantly increases the global DTI for the male mice cohort M1 and the female cohort F1 (two-sample Welch's $t$-test, variability from random halves of the data; see 'Materials and methods' for details of the significant test). In contrast, in the control cohort M4, injecting male mice with BSA either does not significantly change the global DTI or decreases it. Notably, the difference of the DTI is not due to the control group M4 has less mice as subsampling both on the level of the inferred interactions (*Figure 4—figure supplement 1*) and on the level of the mice locations (*Figure 4—figure supplement 2*) give the same DTI for cohorts M1 and F1. This increase of the global DTI is due to the increasing variance of the interaction $J_{ij}$, which is related to more of the negative interactions. Randomly shuffling $J_{ij}$ does not change the global DTI, indicating that no network structure was found that contributes to this global DTI (*Figure 4—figure supplement 3*).

## Discussion

We demonstrate how the joint probability distribution of the mice positions in the Eco-HAB can be used to quantify sociability. By building a pairwise interaction model whose parameters are learned directly from the data, we quantify how much the combined activities of all mice in the cohort are influenced by their individual preferences and how much by the social context. This approach shows that, within the setup of the seminaturalistic Eco-HAB experiments, pairwise interactions between mice are sufficient to describe the statistics of collective behavior in larger groups. Additionally, the pairwise interaction model can capture changes in the social interactions of the network induced by alterations in the neuronal plasticity of the PL in the tested subjects. The Eco-HAB, combined with this analytical approach, provides a toolbox to quantify sociability in mice, which can be applied generally to different mouse strains to study various behavioral phenotypes, including characteristics associated with neurodevelopmental disorders such as autism. Compared to traditional experimental methods like the three-chamber test, our study combines the advantages of an ecologically relevant and automatic experimental apparatus with the powerful tools of statistical inference. The use of statistical inference methods such as maximum entropy models disentangles the effects of individual preferences versus pairwise social interactions in generating the patterns of mouse positions within their territory.

The challenge in studying social behavior lies in finding a balance between being specific enough to capture the properties of sociability while avoiding the loss of generalizability. Including excessive details, such as the classification of precise social behavior among mice, may lead to a more accurate description of the specific mice cohort studied, such as the construction of a precise social network. However, it is difficult to assess comparability across different cohorts of mice. Alternatively, as used in this article, one can construct minimal models and use the ensemble statistics of the models to quantify social properties of a mouse strain without explicitly constructing social networks for each cohort. For example, our study found that the inferred interaction has similar ensemble statistics across four different male cohorts of the same strain but differs across different sexes. This provides evidence to support our argument for a coarse-grained description of mouse social behavior.

Another challenge in studying social behavior lies in the interplay of timescales. We found in this article that in order to gather enough statistics and avoid overfitting, we need to construct probabilistic models using aggregated 5-day data, which poses a challenge to balance model construction with enough data versus studying the temporal evolution of sociability. Is the variability of the social network a true property of the social interaction of the mice cohort, or is it due to variabilities of the inferred model caused by the finite data? To address this question, one needs to consider various timescales. For example, mice–mice interactions occur at a much shorter timescale compared to the timescale of changes in the social network, while in between, there are

the timescales of adaptation to the new environment and the circadian cycle. These issues need to be addressed using a combination of theoretical tools and experimental validation methods in future works.

Additionally, we have simplified our analysis by focusing on a 6-hour time window each day, during which the rate of locomotor activity is most stable. This approach allowed us to circumvent issues related to individual or strain differences in the circadian cycle, such as the observed 'lunch hour' in C57 male mice. One avenue for future research involves reintroducing the circadian cycle as a latent variable to better explain the system. However, caution must be exercised to differentiate between group behavior influenced by the circadian cycle of individual mice and emergent behavior resulting from interactions.

While not the main focus of this article, we tested our methods of quantifying sociability on both female and male mice of the same strain undergoing the same alteration of neuronal plasticity in the PL. Interestingly, while both male and female mice demonstrate an increase of competitive pairwise interactions upon such alteration, they also exhibit many differences in sociability: for example, the individual preferences increase more in male mice after TIMP-1 injection than the female mice.

We will now discuss the relationship between our study and that of *Shemesh et al., 2013* wherein the authors applied a similar approach, investigating the social behavior of groups of four mice in a complex experimental environment using statistical modeling of the joint probability distribution of mice locations. Both Shemesh et al. and us use maximum entropy models to analyze co-localization patterns of a group of mice housed in complex environments. While Shemesh et al. found that triplet interactions are necessary to describe collective behavior, we found that triplet interactions can be predicted by the pairwise model. We suspect the difference in our results could arise from three factors. First, the experimental arena is quite different between the Eco-HAB and what was used in Shemesh et al. – while the Eco-HAB mimics a naturalistic environment with tubes and connected chambers, the experimental arena in Shemesh et al. contains only one chamber, which could cause a difference in how mice interact. Second, even if the interaction patterns do not change, the nature of the data is different: our data is more coarse-grained spatially as the state of each mouse is determined by the large compartment it is in, whereas in Shemesh et al., the location is more precise (e.g., a door, a pillar, etc.). As suggested by a comparison to renormalization theory in physics, at coarser spatial scales, the importance of higher-order interactions is likely to decrease. Finally, our studies include more mice (10–15) compared to Shemesh et al. (4 mice), which may also influence the importance of higher-order interactions. Larger group size also means that including triplet interactions in our model causes overfitting, which restricts the models we have access to pairwise interactions. To further investigate these effects, future experiments in Eco-HAB could include mice cohorts of smaller sizes. More generally, this discrepancy when looking at different choices of variables raises the issue that when studying social behavior of animals in a group it is important to test and compare interaction models with different complexity (e.g., pairwise or with higher-order interactions). Furthermore, since pairwise maximum entropy model is one of the simplest of all maximum entropy models that can describe interactions among individuals, it serves as an excellent starting point to describe collective and social behavior in animals.

How do we move forward, and what is the ideal experiment to study social behaviors? We believe that Eco-HAB offers a balance between a semi-natural environment and controllability, which works well in studying social behavior. One direction for future experimental studies is to focus on the biological function of social interactions. For example, how do mouse cohorts respond to novel odors and transmit information among the cohort? What is the speed of information transmission related to sociability? The current configuration of the Eco-HAB already allows for the introduction of novel odors accessible to all mice, while the next generation of experiments will localize the introduction of information to individuals. From the analysis perspective, as presented in this article, our current model is purely static. Our model describes the joint probability distribution of mice positions within the territory at concurrent time points and does not model the dynamics of the cohort. To take into account the dynamic aspect of social behaviors, such as dominant mice actively chasing others, one will need to build dynamic models of interaction. For example, this can be done by modeling the probability of transitioning to another compartment of each mouse as a function of the history of its previous location and the locations of all other mice (*Chen et al., 2023b*).

## Materials and methods

### Animals

Animals were treated in accordance with the ethical standards of the European Union (directive no. 2010/63/UE) and Polish regulations. All the experiments were preapproved by the Local Ethics Committee no. 1 in Warsaw, Poland. C57BL/6J male and female mice were bred in the Animal House of Nencki Institute of Experimental Biology, Polish Academy of Sciences or Mossakowski Medical Research Centre, Polish Academy of Sciences. The animals entered the experiments when 2–3 months old. They were littermates derived from several breeding pairs. The mice were transferred to the animal room at least 2 weeks before the experiments started and put in groups of 12–15 in one cage (56 cm × 34 cm × 20 cm) with enriched environment (tubes, shelters, nesting materials). They were kept under 12 hours/12 hours light–dark cycle. The cages were cleaned once per week. Five cohorts of mice were used in the analysis of this articl (see details in *Table 1*).

### Exclude inactive and dead mice from analysis

Mouse whose trajectory does not cover all four compartments within the 6-hour period for at least 1 day of the experiment is defined as inactive and excluded from the analysis. Including inactive mice in the maximum entropy model will result in unstable learned parameters, as shown by bootstrapped results. For the same mouse cohort before and after injection of drug (M1, M4, and F1), if a mouse is dead or inactive in either phase of the experiment, its trajectory is masked out from the data for consistency of comparison before and after. Specifically, for cohort F1, mouse number 13 (in the original ordering of the 14 mice) is inactive after the drug application. For cohort M4, mouse numbers 3 and 11 (in the original ordering of the 12 mice) died after surgery, mouse number 9 (in the original ordering) was inactive on the 10th day after drug injection. The total number of mice used in the analysis is given in *Table 1*.

### Longitudinal observation of social structure in the Eco-HAB

Cohorts of mice with the same gender and same strain were placed in the Eco-HAB systems and observed for 10 days, removed from the system to undergo stereotaxic injections with TIMP-1-loaded nanoparticles. After 4–6 days of recovery, the mice were placed back to a cleaned Eco-HAB and observed for 10 days.

### Activity level

The activity level for a given mouse $i$ during a given time period $(t_i, t_f)$ on day $d$ is computed by counting the number of times the mouse passes by any antenna and denoted by $\phi_i^{(d)}(t_i, t_f)$.

Averaging this quantity over all $N$ mice, one obtain the mean activity level for all mice during a given time period. Mathematically, $\overline{\phi}^{(d)}(t_i, t_f) \equiv \sum_i \phi_i^{(d)}(t_i, t_f)/N$. The standard deviation across all days is the day-to-day variability of mean activity level.

Averaging this quantity over all $T$ days, one obtains the mean activity level for each mouse. Mathematically, $\widetilde{\phi}_i(t_i, t_f) = \sum_d \phi_i^{(d)}(t_i, t_f)/T$. The standard deviation across all mice is the mouse-to-mouse variability of the mean activity level.

### Mice location

The raw data consists of time points when mice cross an antenna, as well as the identity of the specific antenna, which are placed at the ends of the four tunnels. The location of a mouse at any given time point is deduced from the most recent time stamps before and after the current time point. For simplicity, for the time points when a mouse is in the tunnel, the location of the mouse is set to be the compartment it will enter. The time resolution is set to 2 seconds as two adjacent time stamps with a separation of less than 2 seconds are likely an artifact of mice sniffing the tunnel and returning to the previous compartment.

### Pseudocounts

Observing the mice for a finite amount of time means sometimes we have the situation where the mice are stuck in the same compartment for the entire 6 hours of observation. This is not common,

but this messes up our statistical inference or model-building procedure. To avoid this situation, we use pseudocounts that smoothen the observed statistics. We define

$$m_{ir} = \frac{1}{T + \lambda} \left( \frac{\lambda}{q} + \sum_{t=1}^{T} \delta_{\sigma_i(t)r} \right) \tag{3}$$

and

$$\Gamma_{ij} = \frac{1}{T + \lambda} \left( \frac{\lambda}{q} + \sum_{t=1}^{T} \delta_{\sigma_i(t)\sigma_j(t)} \right), \tag{4}$$

where $q = 4$ is the number of possible states, $T$ is the total number of time points in the data, and $\lambda$ is the parameter for the pseudocount. In our analysis, after scanning through a range of values for $\lambda$, we set $\lambda = 8$ around which value the outcome remained largely unchanged.

## The probability model
### Gauge fixing for the local field
The probability model is equivalent for the local fields upon a constant, that is, $P(h)$ and $P(h_{ir} + \delta h_i)$ are equivalent. We overcome this redundancy by enforcing the sum of all local fields for each mouse to be zero, that is, $\sum_r h_{ir} = 0$.

### Learning the probability model
We train the model using gradient descent at each learning step $k$ updating the parameters by $J_{ij}^{(k+1)} = J_{ij}^{(k)} - \alpha(c_{ij}^{\text{model}} - c_{ij}^{\text{data}})$ and $h_{ir}^{(k+1)} = h_{ir}^{(k)} - \alpha(m_{ir}^{\text{model}} - m_{ir}^{\text{data}})$, where $\alpha = 0.25 \sim 0.8$ is the step size of learning. The stopping condition is set such that when the difference between the model-predicted correlation and magnetization is less than the data variability, estimated by extrapolation from random halves of the data. In addition, because we are interested in quantifying social properties of mice cohort using the statistics of learned parameters, we add to the stopping condition that the mean and the variation of inferred interaction reach a stable value, with change less than 0.005 over 100 learning steps.

## Computing higher-order correlations
The connected three-point correlation function gives the frequency of finding three mice in the same compartment, subtracting the contributions from the mean and the pairwise correlation. Mathematically,

$$C_{ijk} = \sum_{r=1}^{p} \left\langle (\delta_{\sigma_i r} - \langle \delta_{\sigma_i r} \rangle)(\delta_{\sigma_j r} - \langle \delta_{\sigma_j r} \rangle)(\delta_{\sigma_k r} - \langle \delta_{\sigma_k r} \rangle) \right\rangle \tag{5}$$

If we only subtract the individual preference, then we define

$$C_{ijk}^* = \sum_{r=1}^{p} \langle \delta_{\sigma_i r} \delta_{\sigma_j r} \delta_{\sigma_k r} \rangle - \langle \delta_{\sigma_i r} \rangle \langle \delta_{\sigma_j r} \rangle \langle \delta_{\sigma_k r} \rangle \tag{6}$$

## Comparing in-state probability between model prediction and data
Given time-series data and the inferred joint probability distribution of mice location, we can compare the in-state probability of single mouse, as given by model prediction versus data observation.

More precisely, for each time point $t$, given the inferred compartment preference $h_{ir}$ for mouse $i$, the inferred pairwise interaction $J_{ij}$, and the position of all other mice $j \neq i$, we use the pairwise maximum entropy model (*Equation 2*) to compute the marginal probability of mouse $i$ being in each of the four compartments as the *model-predicted* 'in-state probability', $P_{ir}^{\text{in-state}}(t)$. These model-predicted in-state probabilities are then binned according to their percentiles across all observed time points for each mouse and each box, such that the number of time points in each bin is equal. Then, for all time points that belong to each bin, the frequency count of whether mouse $i$ is actually observed in compartment $r$ is computed as the *observed* in-state probability. Agreement between

the model-predicted and the observed in-state probability across all the bins shows the model is an unbiased estimator, which is what we see after averaging the in-state probability across the four compartments for fixed percentiles in *Figure 2—figure supplement 4*.

## Compute mutual information between single mouse position and the rest of the network

The mutual information between single mouse position and the rest of the network is a measure of collectiveness. For Eco-HAB with four compartments, the mutual information is between 0 and 2 bits. If the mutual information is close to 2 bits, knowing where other mice are is a perfect predictor for the position of single mouse. If the mutual information is close to 0 bits, knowing where other mice are do not help predict the position of the singled-out mouse. The mutual information can be computed as the difference of the entropy of mouse $i$ and the conditional entropy of mouse $i$ with respect to the state of all other mice. Mathematically,

$$I(\sigma_i; \{\sigma_j\}_{j\neq i}) = H(\sigma_i) - \langle H(\sigma_i|\{\sigma_j\}_{j\neq i})\rangle_{\{\sigma_j\}_{j\neq i}}, \tag{7}$$

where the entropy of mouse $i$ is

$$H(\sigma_i) = -\sum_{r=1}^{4} P(\sigma_i = r)\log P(\sigma_i = r) \tag{8}$$

and the conditional entropy is computed using the conditional probability given $\{\sigma_j\}$ and the inferred pairwise data, and averaged over all observed data patterns $\{\sigma_j(t)\}$.

$$
\begin{aligned}
&H\left(\sigma_i|\{\sigma_j\}_{j\neq i}\right)\\
&= -\sum_{\{\sigma_i\}} P\left(\{\sigma_i\}\right)\log P\left(\sigma_i|\{\sigma_j\}_{j\neq i}\right)\\
&= \sum_{\{\sigma_j\}_{j\neq i}} P(\{\sigma_j\}_{j\neq i})\left[-\sum_{\sigma_i} P(\sigma_i|\{\sigma_j\}_{j\neq i})\log P(\sigma_i|\{\sigma_j\}_{j\neq i})\right]\\
&\approx \frac{1}{T}\sum_{t=1}^{T}\left[-\sum_{\sigma_i} P^{(2)}(\sigma_i|\{\sigma_j(t)\}_{j\neq i})\log P^{(2)}(\sigma_i|\{\sigma_j(t)\}_{j\neq i})\right]
\end{aligned}
\tag{9}
$$

To reach the final results, we approximate the ensemble average over all possible mice configurations with a temporal average over all observed mice configuration in the data. We also replace the true underlying conditional probability of $P(\sigma_i|\{\sigma_j\}_{j\neq i})$ with the inferred pairwise probability model $P^{(2)}$.

## Generate error bars using random bootstrapped halves of the data

Error bars of the observed statistics $\mathcal{O}$ (e.g., pairwise correlation, $C_{ij}$, and probability in each compartment, $m_{ir}$), the inferred parameters $\mathcal{P}$ (e.g., pairwise interaction $J_{ij}$ and compartment preference $h_{ir}$), and the subsequent results $\mathcal{R}$ (e.g., the entropy, $S^{(1,2)}$ and $S^{(1)}$, and the DTI $F$) are bootstrap errors generated by repeatedly taking random halves of the data and computing the deviations in the mean. Specifically, each data set (at least 6 hours in duration) is first divided into time bins of 400 seconds. The length of the time bin is chosen such that it is longer than twice the correlation time for each mouse. Then, random halves of the time bins are chosen to compute the observables, as well as used to train a specific pairwise maximum entropy model, which generates a specific set of learned parameters. The deviation across the random halves, $\sigma_{\text{bs}}$, can be extrapolated to the full dataset by $\sigma = \sigma_{\text{bs}}/\sqrt{2}$.

## Test of significance for comparing observables and inferred parameters

To perform significance test across different days of the experiment, we used the Welch's $t$-test for the mean of the inferred interaction $\langle J_{ij}\rangle$, the mean of the food preference $\langle \Delta h_i\rangle$, the mean of the mutual information between single mouse position and the rest of the network given by the inferred pairwise model, and for the global DTI, $F$. We used two-sample $F$-test for the variability of the inferred interaction. Because we are comparing between pairs of the 5-day aggregate data – the first 5 days before drug injection, the last 5 days before drug injection, the first 5 days after drug injection, and the last 5 days after drug injection – we conduct significance tests using Bonferroni corrections. The number

of tests performed for such pairwise comparison is 6. In *Figure 3*, the asterisks encode the following p-values: $*p \leq 0.05/6$; $**p \leq 0.01/6$; $***p \leq 0.001/6$.

For the significance test comparing the global DTI, random halves of the 5-day aggregated data is chosen 10 times, each used to learn the interaction parameters and compute the global DTI. The variance of the global DTI across the 10 random halves is used as variation due to finite amount of data and is adjusted by $\sigma_\theta = \sigma_{\mathrm{rh}}/\sqrt{2}$. Two-tailed tests are performed, and the Bonferroni correction is applied as the total number of tests performed for the pairwise comparisons for the 5-day aggregated data before and after pharmacological intervention is 6. In *Figure 4*, the asterisks encode the following p-values: $*p \leq 0.025/6$; $**p \leq 0.005/6$; $***p \leq 0.0005/6$.

## Maximum entropy model with triplet interactions

To model the joint probability distribution of mice co-localization pattern, one could in principle increase the number of constraints when constructing the maximum entropy model. One example is the triplet correlation, that is, the probability of any triplets of mice are being found in the same box, $\Gamma_{ijk} = \langle \delta_{\sigma_i \sigma_j} \delta_{\sigma_i \sigma_k} \rangle$. The corresponding maximum entropy model has the probability distribution

$$P^{(3)}\left(\sigma\right) \propto e^{\sum_{i,r} h_{ir} \delta_{\sigma_i r} + \sum_{i<j} J_{ij} \delta_{\sigma_i \sigma_j} + \sum_{i<j<k} G_{ijk} \delta_{\sigma_i \sigma_j} \delta_{\sigma_i \sigma_k}},$$

where $h_{ir}$ is the individual preference of mouse $i$ to be in compartment $r$ and $J_{ij}$ is the pairwise interaction between mouse $i$ and mouse $j$ as before, and $G_{ijk}$ is the triplet interaction among mouse $i$, mouse $j$, and mouse $k$. For a cohort with $N$ mice, the numbers of parameters for triplet interactions is $N(N-1)(N-2)/6$. To avoid overfitting, we applied an L2 regularization on the triplet interaction strength, where we now minimize the objective function

$$L^{(3)} = -\frac{1}{T} \sum_{t=1}^{T} \log P^{(3)}\left(\{\sigma_t\}\right) + \frac{1}{2} \beta_G \sum_{ijk} G_{ijk}^2.$$

This is naturally translated to learning steps: $h_{ir}^{(l+1)} = h_{ir}^{(l)} - \alpha_h (m_{ir}^{\mathrm{model}} - m_{ir}^{\mathrm{data}})$, $J_{ij}^{(l+1)} = J_{ij}^{(l)} - \alpha_J (c_{ij}^{\mathrm{model}} - c_{ij}^{\mathrm{data}})$, and $G_{ijk}^{(l+1)} = G_{ijk}^{(l)} - \alpha_G \left( \Gamma_{ijk}^{\mathrm{model}} - \Gamma_{ijk}^{\mathrm{data}} + \beta_G G_{ijk} \right)$. The stopping condition is the same as before, focusing on the difference between the model-predicted correlation and magnetization being less than the data variability, and that the mean and variation of the inferred pairwise interaction reach a stable value.

To test the validity of a triplet interaction model, we varied the regularization strength $\beta_G$. For our largest dataset, the cumulative data consist of all 10 days of the experiment on cohort M1 ($N = 15$), we performed cross-validation by splitting the 6-hour data into six different training-test set combo (5 hours of data in training set and 1 hour in the test set). As shown in *Figure 2—figure supplement 5*, the test set likelihood is found to be maximized when the regularization parameter $\beta_G$ is large, which corresponds to close to zero triplet interactions. This indicates that even for our largest datasets including the triplet interactions in the model has increased the model complexity beyond what the data allows, and the best way to avoid overfitting is for us to not consider triplet interactions. Hence, the interaction models we choose to study in this article is restricted to pairwise interaction models.

## Acknowledgements

This work was partially supported by the European Research Council Consolidator grant no. 724208, Fondation Bettencourt-Schueller, 'BRAINCITY – Centre of Excellence for Neural Plasticity and Brain Disorders' project of the Polish Foundation for Science, and the National Science Center grant 2020/39/D/NZ4/01785.

## Additional information

### Competing interests

Aleksandra M Walczak: Senior editor, eLife. The other authors declare that no competing interests exist.

## Funding

| Funder | Grant reference number | Author |
| --- | --- | --- |
| H2020 European Research Council | CoG 724208 | Aleksandra M Walczak |
| Fondation Bettencourt Schueller | | Thierry Mora |
| Polish Foundation for Science | BRAINCITY | Ewelina Knapska |
| Polish National Science Center | 2020/39/D/NZ4/01785 | Ewelina Knapska |

The funders had no role in study design, data collection and interpretation, or the decision to submit the work for publication.

## Author contributions

Xiaowen Chen, Conceptualization, Resources, Data curation, Software, Formal analysis, Validation, Investigation, Visualization, Methodology, Writing – original draft, Writing – review and editing; Maciej Winiarksi, Resources, Data curation, Investigation; Alicja Puścian, Conceptualization, Resources, Data curation, Validation, Investigation, Visualization, Methodology, Writing – original draft, Writing – review and editing; Ewelina Knapska, Conceptualization, Resources, Data curation, Validation, Visualization, Methodology, Writing – original draft, Project administration, Writing – review and editing; Thierry Mora, Conceptualization, Resources, Formal analysis, Supervision, Funding acquisition, Validation, Investigation, Visualization, Methodology, Writing – original draft, Project administration, Writing – review and editing; Aleksandra M Walczak, Conceptualization, Resources, Formal analysis, Supervision, Funding acquisition, Investigation, Visualization, Methodology, Writing – original draft, Project administration, Writing – review and editing

## Author ORCIDs

Xiaowen Chen ⓘ https://orcid.org/0000-0002-4029-1805
Alicja Puścian ⓘ https://orcid.org/0000-0002-7029-1275
Ewelina Knapska ⓘ https://orcid.org/0000-0001-9319-2176
Thierry Mora ⓘ https://orcid.org/0000-0002-5456-9361
Aleksandra M Walczak ⓘ https://orcid.org/0000-0002-2686-5702

## Ethics

Animals were treated in accordance with the ethical standards of the European Union (directive no. 2010/63/UE) and Polish regulations. All the experiments were pre-approved by the Local Ethics Committee no. 1 in Warsaw, Poland. C57BL/6J male and female mice were bred in the Animal House of Nencki Institute of Experimental Biology, Polish Academy of Sciences or Mossakowski Medical Research Centre, Polish Academy of Sciences.

Reviewer #3 (Public review): https://doi.org/10.7554/eLife.94999.4.sa1
Author response https://doi.org/10.7554/eLife.94999.4.sa2

# Additional files

## Supplementary files

MDAR checklist

## Data availability

All data used in our manuscript and the MATLAB and python code to analyze the data can be found in https://github.com/statbiophys/social_mice, (copy archived at *Chen et al., 2023a*).

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
