## [Editor Report · eLife Assessment]

This **valuable** work investigates the social interactions of mice living together in a system of multiple connected cages. It provides **solid** evidence for a statistical approach capturing changes in social interactions after manipulating prefrontal cortical plasticity. This research will be of broad interest to researchers studying animal social behavior.

---

## [Referee Report · Reviewer #3 (Public review)]

Summary:

Chen et al. present a thorough statistical analysis of social interactions, more precisely, co-occupying the same chamber in the Eco-HAB measurement system. They also test the effect of manipulating the prelimbic cortex by using TIMP-1 that inhibits the MMP-9 matrix metalloproteinase. They conclude that altering neural plasticity in the prelimbic cortex does not eliminate social interactions, but it strongly impacts social information transmission.

Strengths:

The quantitative approach to analyzing social interactions is laudable and the study is interesting. It demonstrates that the Eco-HAB can be used for high throughput, standardized and automated tests of the effects of brain manipulations on social structure in large groups of mice.

Weaknesses:

A demonstration of TIMP-1 impairing neural plasticity specifically in the prelimbic cortex of the treated animals would greatly strengthen the biological conclusions. The Eco-HAB provides coarser spatial information compared to some other approaches, which may influence the conclusions.

---

## [Author Response]

The following is the authors’ response to the previous reviews

**Public Reviews:**

**Reviewer #2 (Public review):**
The authors have constructively responded to previous referee comments and I believe that the manuscript is a useful addition to the literature. I particularly appreciate the quantitative approach to social behavior, but have two cautionary comments.(1) Conceptually it is important to further justify why this particular maximum entropy model is appropriate. Maximum entropy models have been applied across a dizzying array of biological systems, including genes, neurons, the immune system, as well as animal behavior, so would seem quite beneficial to explain the particular benefits here, for mouse social behavior as coarse-grained through the eco-hab chamber occupancy. This would be an excellent chance to amplify what the models can offer for biological understanding, particularly in the realm of social behavior

We thank the reviewer for this comment. Maximum entropy models, along with other statistical inference methods that learn interaction patterns from simultaneously-measured degrees of freedom, help distinguish various types of interactions, e.g. direct vs. indirect interactions among animals, individual preference to food vs. social interaction with pairs. As research on social behavior expands from focusing on pairs of animals to studying groups in (semi-)naturalistic environments, maximum entropy models serve as a crucial link between high-throughput data and the need to identify and distinguish interaction rules. Specifically, among all possible maximum entropy models, the pairwise maximum entropy model is one of the simplest that can describe interactions among individuals, which serves as an excellent starting point to understand collective and social behavior in animals.

Although the Eco-HAB setup currently records spatially coarse-grained data, it still provides more spatial information compared to the traditional three-chamber tests used to assess sociability for rodents. By showing that the maximum entropy model can effectively analyze Eco-HAB data, we hope to highlight its potential in research of social behavior in animals.

To amplify what the models can offer for biological understanding particularly in the realm of social behavior, We have updated the Introduction to add a more logical structure to the need of using maximum entropy models to identify interactions among mice. Additionally, we updated the first paragraph of the Discussion to make it specific that it is the use of maximum entropy models that identifies interaction patterns from the high-throughput data. Finally, we have also added in the Discussion (line 422-425) arguments supporting the specific use of pairwise maximum entropy models to study social behaviors.

(2) Maximum entropy models of even intermediate size systems involve a large number of parameters. The authors are transparent about that limitation here, but I still worry that the conclusion of the sufficiency of pairwise interactions is simply not general, and this may also relate to the differences from previous work. If, as the authors suggest in the discussion, this difference is one of a choice of variables, then that point could be emphasized. The suggestion of a follow up study with a smaller number of mice is excellent.

We thank the reviewer for raising the issue and agree that the caveat of how general pairwise interactions can describe social behavior of animals needs to be discussed. We have added a sentence in the Discussion to point out this important caveat. “More generally, this discrepancy when looking at different choices of variables raises the issue that when studying social behavior of animals in a group, it is important to test and compare interaction models with different complexity (e.g. pairwise or with higher-order interactions).” We have also toned down our conclusion to limit our results of pairwise interactions describing mice co-localization patterns to the data collected in Eco-HAB (also see Reviewer 3 Major Point 2).

**Reviewer #3 (Public review):**
Summary:Chen et al. present a thorough statistical analysis of social interactions, more precisely, co-occupying the same chamber in the Eco-HAB measurement system. They also test the effect of manipulating the prelimbic cortex by using TIMP-1 that inhibits the MMP-9 matrix metalloproteinase. They conclude that altering neural plasticity in the prelimbic cortex does not eliminate social interactions, but it strongly impacts social information transmission.Strengths:The quantitative approach to analyzing social interactions is laudable and the study is interesting. It demonstrates that the Eco-HAB can be used for high throughput, standardized and automated tests of the effects of brain manipulations on social structure in large groups of mice.Weaknesses:A demonstration of TIMP-1 impairing neural plasticity specifically in the prelimbic cortex of the treated animals would greatly strengthen the biological conclusions. The Eco-HAB provides coarser spatial information compared to some other approaches, which may influence the conclusions.
**Recommendations for the authors:**

**Reviewer #3 (Recommendations for the authors):**
Major points(1) Do the Authors have evidence that TIMP-1 was effective, as well as specific to the prelimbic cortex?

We refer to the literature for the effectiveness and specificity of TIMP-1 to the prelimbic cortex.

Specifically, the study by Okulski et al. (Biol. Psychiatry 2007) provides clear evidence that TIMP1 plays a role in synaptic plasticity in the prefrontal cortex. They showed that TIMP-1 is induced in the medial prefrontal cortex (mPFC) following stimulation that triggers late long-term potentiation (LTP), a key model of synaptic plasticity. Overexpression of TIMP-1 in the mPFC blocked the activity of matrix metalloproteinases (MMPs) and prevented the induction of late LTP in vivo. Similar effects were observed with pharmacological inhibition of MMP-9 in vitro, reinforcing the idea that TIMP-1 regulates extracellular proteolysis as part of the plasticity mechanism in the prefrontal cortex. These findings confirm that TIMP-1 is both effective and active in this specific brain region.

Further evidence comes from Puścian et al. (Mol. Psychiatry 2022), who used TIMP-1-loaded nanoparticles to influence neuronal plasticity in the amygdala. They found that TIMP-1 affected MMP expression, LTP, and dendritic morphology, showing its impact on synaptic modifications. More directly relevant, Winiarski et al. (Sci. Adv. 2025) demonstrated that injecting TIMP-1-loaded nanoparticles into the prelimbic cortex altered responses to social stimuli, further supporting the idea that TIMP-1 has region-specific effects on behavioral processes.

We have also updated the main text (page 8, 1st paragraph of “Effect of impairing neuronal plasticity in the PL on subterritory preferences and sociability”) of the manuscript to include the above references.

(2) The Authors seem to suggest that one main reason for the different results compared to Shemesh et al. 2013 was the coarseness of the Eco-HAB data. In this case, I think this conclusion should be toned down because of this significant caveat.

We thank the reviewer for pointing this out, and agree that this caveat and difference should be emphasized. To tone down the conclusion, we have

(1) added details about the Eco-HAB (it being coarse-grained, etc.) in the abstract to tone down the conclusion.

(2) added to the results summary in the Discussion (top of page 12) that the results are “within in the setup of the semi-naturalistic Eco-HAB experiments”

(3) added to the Discussion (page 13) that the different results compared to Shemesh et al 2013 means that general studies of social behavior need to compare models with different levels of complexity (e.g. pairwise vs. higher-order interactions). (Also see Reviewer 2 Comment 2.)

Minor points(1) Please explain what is measured in Fig. 1C (what is on the y axis?).

Figure 1C shows the activity of the mice as measured by the rate of transitions, i.e. the number of times the mice switch boxes during each hour of the day, averaged over all N = 15 mice and T = 10 days (cohort M1). The error bars represent variability of activities across individuals or across days. For mouse-to-mouse variability (blue), we first compute for each mouse its number of transitions averaged over the same hour for all 10 days, then we compute its standard deviation across all 15 mice and plot it as error bars. For day-to-day variability (orange), we first compute for each day the number of transitions for each hour averaged over all mice, then compute its standard deviation across all 10 days as the errorbar. We have added the detailed explanation in the caption of Figure 1C.

(2) In Fig. 3, it would be better to present the control group also in the main figure instead of the supplementary.

We have merged Figure 3 and Figure 3 Supplementary 1 to present the control group also in the main figure.

(3) In Fig. 3 and corresponding supplements, there seems to be a large difference between males and females. I think this would deserve some more discussion.

While not being the main focus of this paper, we agree with the reviewer that the difference between male and female is important and deserves attention in the discussion and also future study. Thus we have added a paragraph in the Discussion (line 394-399, bottom of page 12).